# Quantification of the Quercetin Nanoemulsion Technique Using Various Parameters

**DOI:** 10.3390/molecules28062540

**Published:** 2023-03-10

**Authors:** Manish Kumar Sah, Bibaran Gautam, Krishna Prasad Pokhrel, Lubna Ghani, Ajaya Bhattarai

**Affiliations:** 1Department of Chemistry, Mahendra Morang Adarsh Multiple Campus, Tribhuvan University, Biratnagar 56613, Nepal; 2Central Department of Chemistry, Tribhuvan University Campus, Kathmandu 44618, Nepal; 3Department of Chemistry, Women University of Azad Jammu and Kashmir, Bagh 12500, Pakistan; 4Department of Chemistry, Indian Institute of Technology, Chennai 600036, India

**Keywords:** quercetin, nanoemulsions, lipophilicity, polyphenolic flavonol

## Abstract

Natural antioxidant polyphenolic compounds obtained from different plants are considered antioxidants for curing various chronic pathological diseases such as cardiovascular disorders and cancer. Quercetin (a polyphenolic flavonol) has attracted much attention from dietitians and medicinal chemists due to its wide variety of pharmacological activities, including anti-diabetic, anti-hypertensive, anti-carcinogenic, anti-asthmatic, anti-viral, and antioxidant activities. Furthermore, structurally, it is well suited to stabilize emulsions. The present review depicts the important role of the quercetin nanoemulsion technique, used to enhance the solubility of target materials both in vivo and in vitro as well as to decrease the risk of degradation and metabolism of drugs. Researchers have used cryo-TEM to study the morphology of quercetin nanoemulsions. The effects of various parameters such as pH, salts, and solvent concentration on quercetin nanoemulsion have been investigated for quercetin nanoemulsion. Many studies have used UV–Vis spectroscopy and HPLC for the characterization of these particles such as solubility, stability, and encapsulating efficiency.

## 1. Introduction

Quercetin (3,3′,4′,5,7-pentahydroxy-flavone) (QUE) is known as a polyphenolic bioflavonoid and is found in green tea, onions, apples, and many other fruits [1]. The radical-scavenging efficiency of flavonoids is due to their phenolic hydroxyl groups and their anti-inflammatory effects. Quercetin is considered the most significant compound, acting as an anti-allergic, anti-inflammatory, antiviral, anti-obesity, and anti-carcinogenic compound [2,3,4]. One of the information sources obtained from the US Department of Health and Human Services reveals that the average daily intake of QUE is about 25 mg in humans [5]. QUE was able to inhibit the oxidation of LDL in vitro at a concentration as low as 0.25 μmol/L [6,7]. Therefore, quercetin might contribute to the prevention of cardiovascular disease [8]. However, to induce these health effects in humans, quercetin must enter the systemic circulation. QUE in foods is bound to sugars, with the most common compounds being quercetin-3-O-glucoside or aglycone and the bioavailability of these various compounds is affected by their sugar moiety. Moreover, QUE is effective against malignant tumors of the prostate, liver, lung, and breast [9]. However, the high lipophilicity of QUE decreases its oral bioavailability, creating a desire in the minds of researchers to design an alternate path that ensures QUE’s use in many pathological contexts. Recently, it was reported that the synergic effect of QUE in combination with cisplatin is used for cancer treatment [10].

QUE is poorly soluble in water, with an oral bioavailability of about 4%. The emulsion is formed by mixing two immiscible liquids, which are then converted into a single phase by using emulsifying agents such as surfactants and cosurfactants that are thermodynamically stable. There are three types of Nes: oil-in-water (O/W) Nes, in which oil is diffused in a continuous aqueous phase; water-in-oil (W/O) Nes, in which water is dispersed in a continuous oil phase; and bicontinuous NEs. Although NEs can take on many different forms, they are frequently seen as swollen, spherical micelles or bicontinuous structures [11]. The two kinds of emulsion techniques, i.e., microemulsion and nanoemulsion, were used to increase solubility for the delivery of target materials both in vivo and in vitro, as well as to decrease the degradation and metabolism of drugs.

NPs are tiny materials, having sizes ranging from 1 to 100 nm. Due to their various applications, they are growing rapidly in the fields of Nanoscience and Nanotechnology. Recently, researchers have been focusing much of their efforts on the synthesis and applications of various nanomaterials due to their potential applications in science and industry [12]. Two approaches are used to synthesize these nanoparticles, i.e., the bottom-up approach and the top-down approach. Among the methods used, recently, nanomaterials were synthesized by physical vapor deposition, chemical vapor deposition, electrospinning, 3D printing, biological synthesis, and supercritical fluid. In order to improve the synthesis efficiency, these methods were mingled with other methods [13,14]. Their outstanding features such as large surface area, surface functionalization, and tunable porosity explain why nanomaterials are the perfect candidates in the biomedical sector for the production of tissue-engineered scaffolds (e.g., blood vessels or bone), drug delivery systems (gene therapy, cancer treatments, or drugs for chronic respiratory infections), chemical sensors [15,16], biosensors [17], and wound dressings [18]. The chemical composition, size, shape, surface charge, area, and entry route of nanoparticles can influence their biological activities and effect on the body.

Controllable particle sizes of nanoparticles were synthesized by using two dispersion methods, either nanoemulsion or microemulsion. Nanoemulsions consist of very small emulsion droplets, commonly oil droplets (continuous phase) in water (dispersed phase), exhibiting sizes lower than ∼100 nm. They are thermodynamically unstable and require a cosurfactant for their stabilization due to high energy. Nanoemulsions are only formed if surfactants are first mixed within the oily phase [19]. They are created to improve the delivery of active pharmaceutical constituents to cells. Microemulsions are considered thermodynamically stable dispersions of a mean droplet size of approximately 100–400 nm. QUE microemulsion is formed by spontaneous emulsification in which oil, surfactant, and co-surfactant mixtures are slowly poured together [10]. Microemulsions exhibit a large range of structures having different types of nanometric-scaled morphologies. The main difference between nanoemulsions and microemulsions is the influence of the order in which the different compounds are mixed during formulation [19].

QUE microemulsion is critical for increasing oral bioavailability and evaluating its potential clinical utility against certain inflammatory and allergic diseases. One of the studies revealed that QUE nanoparticles (nanoemulsions) can reduce the risk of injuries caused by intestinal mucositis caused by methotrexate. Another study examined the use of silica nanoparticles to enhance the quality of chemotherapy in gastric cancer patients by loading QUE and doxorubicin [20]. QUE microemulsions can improve the low solubility of QUE in aqueous and organic vesicles. QUE microemulsions were used for the skin delivery of quercetin, which exerts antioxidative effects [21,22]. Lipid-based delivery methods are highly suitable formulation strategies to improve the solubility and bioavailability of hydrophobic components [23].

Vicentini et al. observed a photoprotective effect of w/o microemulsion and quercetin as a counter to dermal damage. It can modulate NF-ĸB activation, as observed via many inductors. During in vivo studies, it is well known that microemulsion-quercetin (MEQUE) topical treatment can decrease the behavior of UV-B-exposed skin matrix metalloproteinases (MMPs) via MMP inhibition. When the experiment was carried out on hairless mice with MEQUE, levels of glutathione (GSH) decreased. MEQUE formulation acts as a photo-chemoprotective agent on human skin, which protects skin from UV radiation [24,25]. Savale et al. made microemulsion quercetin (MEQUE) for treating brain tumors through the intranasal pathway. It is known that intranasal drug delivery is known as effective for neurotherapeutic purposes. QUE can inhibit angiogenesis and decrease the rate of formation of new blood cells within blood vessels, which creates tumor growth [10]

## 2. Effects of External Parameters on Quercetin Nanoemulsion

### 2.1. Effect of Temperature

Dario et al. illustrated that in the experiment, throughout the testing period, the amount of quercetin in the nanoemulsion stored in the dark at ambient temperature (22.0 ± 2.0 °C) or in a refrigerator (5.0 ± 2.0 °C) remained consistent. Only at high-temperature storage conditions (45.0 ± 2.0 °C) was degradative oxidation seen. This was demonstrated by a 90-day decline of 9% in quercetin concentration, a 20% drop in pH, and a 25% decline in antioxidant activity [26].

It has been examined how temperature affects the morphology of quercetin dihydrate. We can see that as the temperature rises to 80 °C and then even higher to 140 °C, there is a noticeable drop in the size of the crystals. As was already indicated, this decrease in particle size may be related to a rise in quercetin dehydrate water solubility with temperature. Additionally, it was noted that as the temperature rises, the quercetin dihydrate crystals appear to group together into a compact configuration. At 140 °C, this effect is extremely strong. The dehydration of quercetin at temperatures above 100 °C is the cause of this crystallization, as shown in Figure 1 [27].

Thermo-gravimetric analyses (TGA) revealed that between 94 and 137 °C, quercetin had a first mass loss (around 6.5%), which was likely caused by the sample’s loss of water. This finding demonstrates that, while being lipophilic, quercetin exhibits a degree of crystal lattice hydration in the solid state. The second stage of mass loss had a mass loss of 27.3% and ran from 240 °C to 385 °C. A third mass loss with a loss of 38% happened in the 386–900 °C range. At 900 °C, the remaining mass was 28.2%. Three breakdown phases were visible in the TGA profile of the binary physical mixture of curcumin and quercetin (1:1). Due to the inclusion of quercetin hydrates in the combination, the initial mass loss was the release of water molecules; this mass loss occurred between 72 and 127 °C and had a lower percentage than that of pure quercetin. The second and third mass losses are in line with the thermal breakdown of the compounds (curcumin and quercetin) at the highest temperatures, which has been observed for pure compounds as well (Figure 2) [28].

Based on particle size, PDI (polydispersity index), and zeta potential for a period of three months at 21 °C and 37 °C, the stability of the quercetin-loaded NEs in long-term storage was examined by Son. H.Y et al. The nanoemulsions particle sizes were 192–282 nm and 178–277 nm, respectively, for particles stored at 21 and 37 °C for three months. The PDIs at both storage temperatures varied from 0.2 to 0.4. At 21 °C, the range of zeta potentials was −42.8 ~ −69.1 mV; at 37 °C, it was −41 ~ −63 mV. This is in line with a study that found quercetin-loaded casein nanoemulsions become unstable when stored for an extended period of time and at high temperatures (above 55 °C) [29].

Overall, it can be seen that temperature has a significant effect on the stability and formulation of quercetin nanoemulsion. The nanoemulsion was generally stable at ambient temperatures, whereas with the increase in temperature, it started to degrade due to oxidative reactions. As the temperature increases to a certain level, the nanoemulsion starts to lose a certain fraction of its mass.

### 2.2. Effect of Solvent

Due to quercetin’s weak solubility in aqueous solutions and low bioavailability, consuming meals or supplements containing the flavonoid may not be sufficient. Therefore, the creation of formulations for quercetin that can increase its water solubility and consequently boost its bioavailability and biological activity is required. The ethyl acetate-water emulsion method is useful because it is a solvent with minimal toxicity. Due to this reason, ethyl acetate has been chosen as the organic solvent. Moreover, using it as a flavoring ingredient is safe. In this study, quercetin was encapsulated using modified n-octenyl succinate anhydride (OSA) starch, soybean lecithin, and barley-glucan, which are three natural surfactants [30].

Using a continuous flow type apparatus, the aqueous solubility of 3,3′,4′,5,7-pentahydroxyflavone (quercetin) and its dihydrate was determined between 25 and 140 °C. The subcritical water flows at 0.1, 0.2, and 0.5 mL/min was studied to check it effect on thermal degradation and quercetin solubility at temperatures above 100 °C. Up to 80 °C, quercetin dihydrate’s water solubility was comparable to that of anhydrous quercetin. The aqueous solubility of quercetin dihydrate was 1.5–2.5 times greater than that of anhydrous quercetin at temperatures more than or equal to 100 °C as represented in Figure 3 [27].

According to the graph, it can be seen that temperature has an exponential effect on anhydrous and dehydrated quercetin’s water solubility. At 25 °C, it was discovered that the difference between the water solubilities of anhydrous quercetin and quercetin dihydrate was statistically negligible (*p* = 0.1678). The two compounds’ aqueous solubilities, measured in grams per liter of solvent, were found to be the same up to a temperature of about 80 °C, beyond which quercetin dihydrate’s solubility in water was discovered to be nearly double that of anhydrous quercetin [27].

Sandip et al., in their research work, used PLGA (poly(D, L-lactide-co-glycolide)) nanoparticles as a drug delivery mechanism where quercetin was the active ingredient in the medication. The solvent effect for acetone and the chloroform/methanol mixture was seen during the solvent evaporation and nanoparticle production processes. Eventually, this ideal organic was chosen as the solvent for creating the hydrophobic drug-loaded polymer nanoparticles (quercetin). From these observations, it was determined that crystals formed because quercetin has very poor water solubility. This is consistent with reports that when quercetin was dissolved in methanol and introduced to water, massive precipitates (average particle diameter > 1 m) with a negative surface charge (17 mV) formed. This current study shows how choosing an appropriate and efficient organic solvent can enhance trapping and ensure the successful production of hydrophobic drug-loaded polymeric nanoparticles [31].

### 2.3. Effect of pH

Nanoemulsions exhibited opaque, undispersed, flocculated, clumped, and separated patterns at pH ranges of 4.0 to 6.0. At pH 6–7, however, no flocculation was seen, indicating that these emulsions were reasonably resistant to droplet aggregation. Additionally, nanoemulsions created at pH levels between 6.5 and 9.0 showed transparency and dispersibility without clumping, flocculation, separation, or opacity, indicating high stability. Further research was conducted using quercetin-loaded nanoemulsions to examine the shape, encapsulation effectiveness, particle size, zeta potential, and long-term physical stability after 3-month storage [29]. There was evidence of monomodal particle size in the 207–289 nm range. Quercetin-loaded nanoemulsions at pH ranges between 4.0 and 6.0 had substantially bigger particle sizes than those at pH ranges between 6.5 and 9.0. Droplet aggregation development and phase separation made the nanoemulsions particularly unstable at low pH. Our findings show that nano-quercetin, generated at P^H^ levels over 6.0, can be reasonably stable in terms of droplet development [32,33]. The allowable PDIs for the quercetin-loaded oil-in-water nanoemulsions were less than 0.47. The relatively low PDI values for pH values between 6.5 and 9.0 suggest that these pH values produce monodisperse systems, which lead to more stable nano-delivery systems in terms of unified droplet distribution.

Figure 4 illustrates that the sodium alginate and lecithin aggregation and gelation at pH 4 caused the quercetin-loaded nanoemulsion to tend to clump, whereas pH levels 5, 6, 7, 8, and 9 caused spherical or oval morphologies and droplet-type nanoemulsion structures.

Even at neutral pH, quercetin undergoes significant chemical modification on metal nanoparticles, primarily involving the dimerization and potential oligomerization of the flavonoid on the surface, as shown by the SERS spectra of quercetin on Ag NPs measured at various pH levels, ranging from the acidic to the alkaline regions. A general weakening of the spectrum intensity is observed with rising pH. This phenomenon is linked to QUE’s ionization during deprotonation at an alkaline pH, which causes the appearance of negative charges in QUE, as well as a rising concentration of hydroxyl ions, which tend to be tightly bonded to metal surfaces [34,35].

One of the studies on rice bran protein-based quercetin-loaded nanoemulsions (QUE-RBP NEs) showed good stability in alkaline environments and with low salt ion concentrations. With an increase in pH, the nanoemulsions’ droplet size shrank while the potential trended lower. This may be because the protein is more soluble the further the pH is from the isoelectric point of RBP (pI = 4.7), and electrostatic repulsion increases as the net charge increases, improving the stability of the nanoemulsion. As the pH rose from 4 to 9, there were reductions in the zeta potential and the average size of the emulsion’s droplets, which are stabilized by various proteins. The protein interaction reduces with increasing pH levels, promoting an increase in protein solubility and molecular flexibility and improving the proteins’ ability to form emulsions [36].

Despite quercetin’s potential as an anticancer medication, its limited solubility and brief biological half-life make long-term targeted delivery difficult. Therefore, it is essential to provide a platform with stimuli-responsive properties for targeted distribution, improve its loading, and lengthen the release time. To improve the loading and sustained release of quercetin while using nanoemulsions, Samadi et al. made a pH-responsive nanocomposite enclosed in nanoemulsions in 2021. As a result of improved loading efficiency and pH-responsive release behavior, HAp nanoparticles are added to nanocomposites. Due to their interactions with drugs and polymers in the nanocomposite structure, Hydroxyapatite (HAp) nanoparticles are used to control the drug’s burst release. This allows the drug to be retained at pH 7.4 while being released at the tumor location with a lower pH [37].

Various studies showed that pH has a great effect on the stability of quercetin nanoemulsion, which means that at a low pH (an acidic medium), it loses its stability and starts to go into a state of opaqueness, non-dispersion, and flocculation; however, as the pH tends to rise, the nanoemulsion starts to lose its turbidity and forms a transparent compound by dissolving in an alkaline solution.

### 2.4. Effect of Salt

The nanoemulsion’s zeta potential and stability coefficient dropped while the droplet size of the QUE-RBP NEs and the PDI steadily grew as the salt ion concentration rose. When the salt ions increased the stability of the nanoemulsion, it weakened, and droplet flocculation could be seen in the nanoemulsion’s microstructure. The promotion of van der Waals forces between the emulsified oil droplets, leading to emulsion aggregation and flocculation, may be the cause of this unfavorable effect of the increased salt concentration [36].

Iqbal et al., in their work, observed that the d_av_ of nanoemulsions was marginally raised by the addition of salt (NaCl). Overall, none of the formulated emulsions’ d_av_ significantly changed. However, the strength of the negative charge decreased in all of the created emulsions as the salt content increased. The drop in charge may be caused by electrostatic screening effects or by a weakening of the electrostatic forces in the nanoemulsions, which prevent each nanoparticle from aggregating and flocculating [38]. As the concentration of salt increases, the stability of quercetin nanoemulsion starts to decrease, and flocculation of nanoemulsion is seen, which is mainly due to the increase in van der Waals’s force of attraction between the emulsified oil droplets.

From various studies, we have concluded that nanoemulsion is generally stable at ambient temperatures (22.0 ± 2.0 °C) or even low temperatures (5.0 ± 2.0 °C), whereas, with an increase in temperature, it starts to degrade due to oxidative reactions. Additionally, it was noted that as the temperature rises, the quercetin dihydrate crystals appear to group together into a compact configuration. Quercetin is almost insoluble in water; due to this, its potential bioactive benefits are not directly consumed by the body. The solubility of quercetin is also greatly affected by temperature; with a temperature rise, quercetin becomes soluble in an aqueous solvent. At 25 °C, it was discovered that the variation within water solubilities of anhydrous quercetin and quercetin dihydrate was statistically negligible. Whereas in non-polar solvents, say ethyl acetate, methanol is more soluble. In contrast, when the pH tends to rise, the nanoemulsion starts to lose its turbidity and forms a transparent compound by dissolving in an alkaline solution. At a low pH below 4 (an acidic medium), it loses its stability and transitions to a condition of opaqueness, non-dispersion, and flocculation. The stability of the quercetin nanoemulsion begins to decline as the salt concentration rises, and flocculation of the nanoemulsion is observed, which is primarily caused by an increase in the van der Waals force of attraction between the emulsified oil droplets.

## 3. Characterization of Quercetin Nanoemulsion

### 3.1. Nuclear Magnetic Resonance (NMR)

NMR is a powerful analytical technique that can provide detailed information about the molecular structure, dynamics, and interactions of compounds in a sample. Measuring the variation in the quercetin molecules’ diffusion coefficients is one method to use NMR to characterize quercetin nanoemulsions. The size and shape of the particles in the emulsion can be determined by the diffusion coefficient, which estimates how quickly molecules travel in a solution. Table 1 displays the relative Diffusion Coefficient of water. The research used NMR experiments to pinpoint the location of the encapsulated quercetin. The outcomes, though, fell short of meeting this objective. However, the fact that the water had a relative diffusion coefficient that was far greater than that of the oil did give additional proof that the formulations did include discrete particle structures [39].

The aldehyde and aromatic protons created distinct signals at 10.52 (s, 1H) and 7.61–8.04 (m, 5H), respectively, when CFQ(2-chloro-3-formylquinoline) was made from acetanilide utilizing a Vilsmeier–Haack reaction. The proposed structure is also supported by the appearance of signals in the 1H NMR spectrum of FQO(3-formylquinolin-2(1H)-one) at 10.27–10.48 (brs, 1H), 10.63 (s, 1H), and 7.72–8.76 (m, 5H) caused by the NH, aldehyde, and aromatic protons. In this case, the 1H NMR spectrum was used to identify the characteristic signals of CFQ and FQO, which confirmed the successful synthesis of the modifying agents. This information is crucial for preparing chitosan-quinoline nanoparticles and for the characterization of the resulting nanoemulsion. Overall, the characterization of quercetin nanoemulsion using NMR provides valuable information about the chemical structure of the modifying agents and their role in preparing chitosan-quinoline nanoparticles [40].

### 3.2. TEM

A beam of electrons is transmitted through a material to create a picture in the microscopy technique known as transmission electron microscopy (TEM). The specimen is typically a suspension on a grid or an ultrathin slice that is no more than 100 nm thick. As the beam passes through the specimen, a picture is created as a result of the electrons’ interactions with it. Following that, the picture is enlarged and focused onto an imaging device, such as a fluorescent screen, a sheet of film, or a sensor such as a scintillator linked to a charge-coupled device.

Following this methodology, the droplet size, surface morphology, and structure of the quercetin particle in the oil-in-water nanoemulsion were characterized by different researchers using high-resolution transmission electron microscopy (TEM). According to the findings of the TEM, the quercetin-loaded nanoemulsion droplets had a limited size distribution and a diameter of 50 nm. They were spherical and uniformly spread, as shown in Figure 1. Additionally, no evidence of precipitation could have compromised the stability of the nanoemulsions [41].

Arbain, N. H et al. have conducted an investigation on a system containing aerosol nanoemulsion with quercetin to determine the shape and size of the nanoemulsion droplets. The TEM image detected a spherical shape of the quercetin (Figure 5), which was in accordance with the DLS of the zeta sizer analysis [42]. Similarly, the globules in the morphological studies of the gel-based nanoemulsion of quercetin were discovered to be well-spaced and of consistent sizes by Gokhale et al. (Figure 6). Additionally, they were non-aggregated, spherical, and had a smooth, flexible border, demonstrating their resilience against Oswald ripening due to the globular collapse [43].

Furthermore, Tran et al. have studied the TEM images (Figure 7) of ideal blank nanoemulsion and quercetin nanoemulsion 72 h after dilution in distilled water. After quercetin inclusion, spherical nanoemulsions developed without experiencing morphological changes. Additionally, there was no evidence of quercetin precipitation in the TEM pictures, indicating that the produced nanoemulsions were stable [44]. In addition, the quercetin droplets in the bioactive cationic nanoemulsion were seen by Dario et al. using cryo-TEM (Figure 8) and were found to be spherical with a diameter of 20–30 nm [45].

TEM was used to detect the shape and morphology of the quercetin nanoemulsion, and it was found that the nanoemulsion was smooth and spherical with varying diameters.

### 3.3. FTIR

Infrared spectroscopy using the Fourier transform (FTIR) converts the interferogram’s raw data into the real spectrum using mathematics. The infrared spectrum of transmission or absorption of a fuel sample is obtained using the FTIR method. The presence of both organic and inorganic chemicals in the sample is determined by FTIR. The exact chemical groups predominating in the sample will be identified using spectrum data in the automated program of spectroscopy, depending on the infrared absorption frequency range of 600–4000 cm^−1^ [46,47]. **FTIR frequency ranges and functional groups present in QUE is given in**
Table 2. 

### 3.4. UV–Vis

Similar to FTIR, UV–Vis spectroscopy is a method that can be used to identify pure pharmacological molecules. Many compounds have chromophores, which can absorb certain visible or ultraviolet light wavelengths. The absorption of spectra produced from various samples at specific wavelengths can be directly correlated with the concentration of the sample by applying the Beer–Lambert law.

The UV–Vis spectroscopy of the quercetin nanoemulsion exhibits two prominent absorption bands: band I at 370 nm and band II at 260 nm, which correspond to the B-ring (cinnamoyl system) absorption and the A-ring (benzoyl system) absorption, respectively. Even though the absorption of the cinnamoyl system was decreased by UV light, the quercetin included in the nanoemulsion was very photostable because only 50% of it was damaged after exposure to a substantial radiation dose (70 kJ/cm^2^). UV–Vis spectroscopy can identify the quercetin forms (cationic, neutral, mono-, di-, and tri-anionic) predominant in the nanoemulsion. There are at least three distinct forms of quercetin, including one anionic form and two tautomers of the neutral form [26].

The effectiveness of drug loading and entrapment efficiency for quercetin can be determined using UV–Vis spectroscopy. Chen-yu et al. optimized quercetin nanoemulsion formulation and yielded an average entrapment efficiency of 89.95 ± 0.16% and an average drug loading of 3.05 ± 0.01%. Additionally, the system could undergo autoxidation in a weakly alkaline environment, and the end product of autoxidation exhibited 325 nm UV absorbance [50].

To calculate the solubility, quercetin was added to 2.0 g of different oils, surfactants, and co-surfactants, and the concentration of the quercetin was calculated using UV–Vis spectroscopy. Figure 9 summarizes the solubility of quercetin in surfactants, co-surfactants, and various oils. The solubility of quercetin was found to be at its highest among the several excipients investigated in TRANS HP, CR RH 40, and CAP MCM NF. Similarly, UV–Vis spectroscopy can be used to investigate the in vitro drug release. Pure quercetin released its drug after being suspended at a rate of 18.12%. However, quercetin nanoemulsion released its drug at a rate of 58.82% after 48 hr. Quercetin was released from nanoemulsions, initially in bursts over the first 10 hrr, followed by steady releases for the next 48 hr. The outcomes for quercetin nanoemulsion demonstrated that the composition could increase medication penetration throughout cell membranes and increase quercetin bioavailability [46].

Tran et al. also studied the solubility of quercetin in different solutions and the quantification of quercetin content using UV–Vis spectroscopy. Different oils, surfactants, and cosurfactants were tested for quercetin solubility (Figure 10). The solubility of quercetin in castor oil, CapmulR MCM, and CapryolR was higher than in labrafil, miglyol 812, and soybean oil. Their effectiveness determined the choice of surfactant in emulsifying rather than by their capacity to solubilize quercetin [44]. Similarly, Silva et al. also studied the concentration of quercetin in the oil-water nanoemulsion using a UV–Vis spectrophotometer [51].

UV–Vis spectroscopy is used to detect the content of quercetin; thus, it can be indirectly used to determine solubility, drug loading, and entrapment efficiency, as well as the different forms of quercetin.

### 3.5. HPLC

The analytical chemistry method of high-performance liquid chromatography (HPLC), formerly known as high-pressure liquid chromatography, is used to separate, recognize, and quantify each component in a mixture. It uses pumps to move a column of solid adsorbent material through a pressured liquid solvent containing the sample combination. As the components flow out of the column, the components separate because each component in the sample interacts slightly differently with the adsorbent material, resulting in varying flow rates for the various components.

The HPLC technique can be used to determine the concentration of quercetin In the nanoemulsion and determine the system’s robustness. The ability of an analytical method to withstand alterations brought on by minute changes in parameter circumstances is known as robustness. Standard solutions (2.5 g/mL) were analyzed in the HPLC method to determine the robustness of the chromatographic method, while minor adjustments were made to the conditions of the procedure, such as pH, mobile phase composition, and column temperature. The composition showed an amount of about 0.71 ± 0.08 mg/mL of quercetin that encapsulates 94.66 ± 10.6% of QUE with an entrapment efficiency of > 99% [28].

Jayanti et al. also studied the % entrapment efficiency (% EE) and quercetin content of the nanoemulsion for rheumatoid arthritis using HPLC. The resultant batch was chosen to create a gel formulation because the drug concentration of the nanoemulsion was determined to be 95.65 ± 0.14% and the improved entrapment effectiveness of the nanoemulsion was determined to be 94.65 ± 0.14%. Furthermore, HPLC was also used to examine the ex vivo permeation studies, and it found that, compared to free quercetin gel, the medication from quercetin nanoemulsion gel penetrated more quickly. The nanoemulsion has a tiny droplet size (200 nm) that allows the medication to pass the protective skin barrier and quickly penetrate the stratum corneum, considerably enhancing the rate of penetration [45].

The stability of quercetin in both negatively and positively charged nanoemulsions maintained in refrigerators (2–8 °C) for 30 and 120 days can also be determined using HPLC, and it was found that the sample remained stable for more than 60 days. Similarly, the HPLC was also used to determine the entrapment efficiency and drug content, where it was demonstrated that quercetin could be encapsulated at a concentration of 0.75 mg/mL in a drug-loaded nanoemulsion with entrapment efficiency ratings of more than 99%, as shown in Table 3 [52].

For the synthesis of a delivery solution containing a nanostructured lipid carrier with quercetin, HPLC was used for the analysis of in vitro skin penetration in an in vivo permeation study in mice. The accumulated quantities of quercetin at various times in the receptor solution from in vitro tests are displayed in Figure 11. The two solutions of quercetin permeation exhibited zero-order release kinetics. At 12 h after dosage, the total quantities of quercetin in the nanostructured lipid carrier and propylene glycol solution were 31.89 ± 3.15 µgcm^2^ and 9.00 ± 1.19 µgcm^2^, respectively. This means that by 12 h, the total quantity of quercetin from the nanostructured lipid carrier that had penetrated the mouse skin was greater than 2.54 times that of the propylene glycol solution [50].

HPLC was used in the in vitro permeation tests via a parallel artificial intestinal membrane to examine the effectiveness of the treatment against obesity in mice. When compared to the quercetin solution employed as the control, a series of quercetin nanoemulsions demonstrated significantly higher levels of permeability in the artificial intestinal membrane permeability testing. When compared to an aqueous dispersion of quercetin, the permeability of quercetin rose from 110 to 180 times, depending on the nanoemulsion. This increase in permeability may have been brought on by the surfactant content rising from 12.5% to 37.5%. This is most probably caused by the droplet size nanoemulsion that is increased with poor solubilizing action of the lowered S_mix_(surfactant and cosurfactant combination) [41].

## 4. Conclusions

Quercetin is a potent molecule that can be used to cure various health-related issues and manifests antioxidant properties both in vivo and in vitro. Due to its low water solubility and poor bioavailability, nano-emulsion delivery systems possess an advantage over other systems in terms of solubility and degradation. The morphology of the quercetin nanoemulsion determined using TEM revealed that the nanoemulsion was smooth and spherical with a range of sizes. Both UV–Vis spectroscopy and HPLC were very useful to determine various properties of quercetin nanoemulsion, such as their solubility, stability, encapsulating efficiency, etc. Nanoemulsion is generally stable at an ambient temperature (22.0 ± 2.0 °C or even low temperature (5.0 ± 2.0 °C), whereas, with the increase in temperature, it starts to degrade due to oxidative reactions. With the increase in pH, nanoemulsion starts to lose its turbidity and forms a transparent compound by dissolving in an alkaline solution. At low pH (below 4) it loses its stability and transitions to a condition of opaque, undispersed, and flocculation. The stability of the quercetin nanoemulsion begins to decline as the salt concentration rises, and flocculation of the nanoemulsion is observed, which is primarily caused by an increase in the van der Waals force of attraction between the emulsified oil droplets.

## Figures and Tables

**Figure 1 molecules-28-02540-f001:**
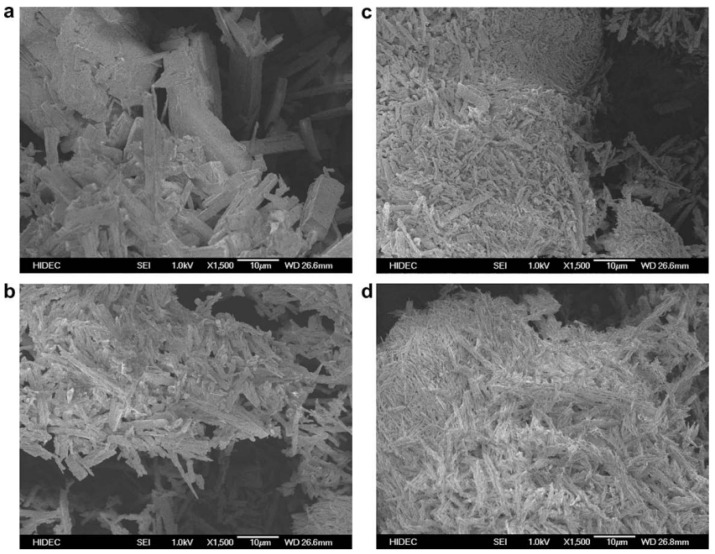
Scanning electron microscopy (SEM) images of anhydrous quercetin at 25 °C (**a**); quercetin dihydrate at 25 °C (**b**); quercetin dihydrate at 80 °C (**c**); and quercetin dihydrate at 140 °C (**d**) [27]. “Reproduced with permission from reference”.

**Figure 2 molecules-28-02540-f002:**
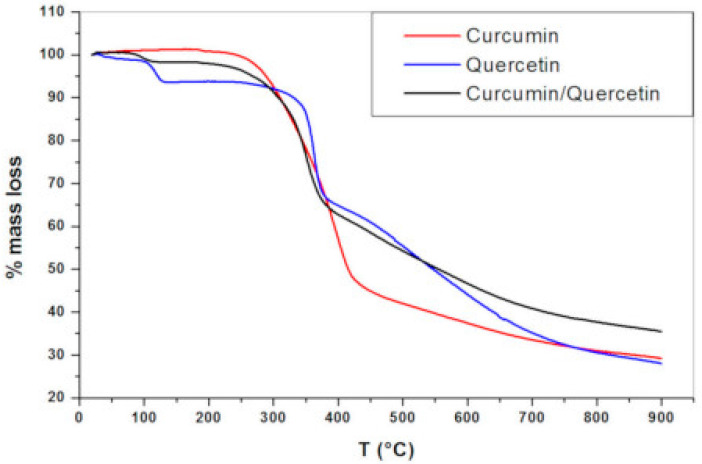
Analysis curve of thermogravimetric testing [28].

**Figure 3 molecules-28-02540-f003:**
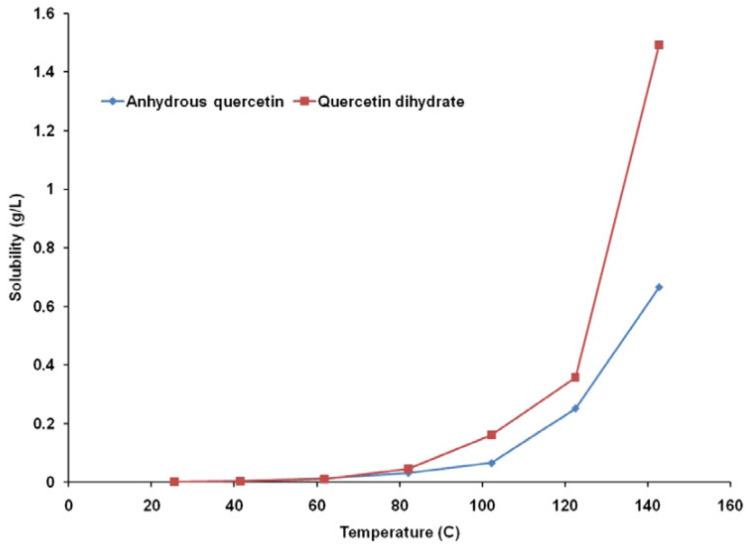
Temperature dependence of quercetin solubility [27]. “Reproduced with permission from reference”.

**Figure 4 molecules-28-02540-f004:**
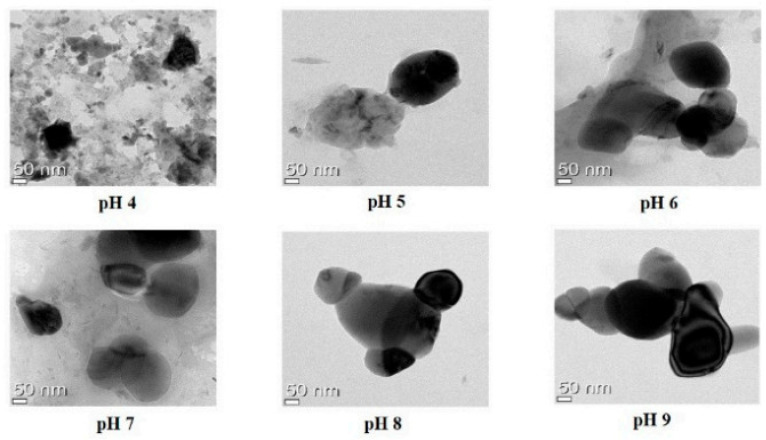
TEM images of quercetin-loaded with different pH levels [29]. “Reproduced with permission from reference”.

**Figure 5 molecules-28-02540-f005:**
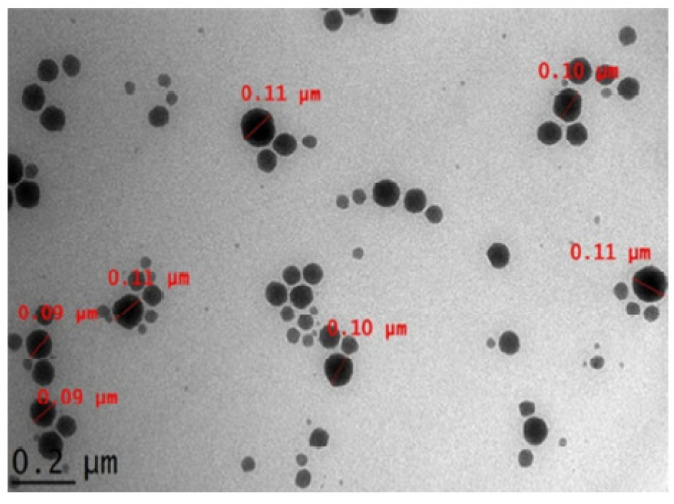
Quercetin-loaded palm-based nanoemulsion in TEM image [42]. “Reproduced with permission from reference”.

**Figure 6 molecules-28-02540-f006:**
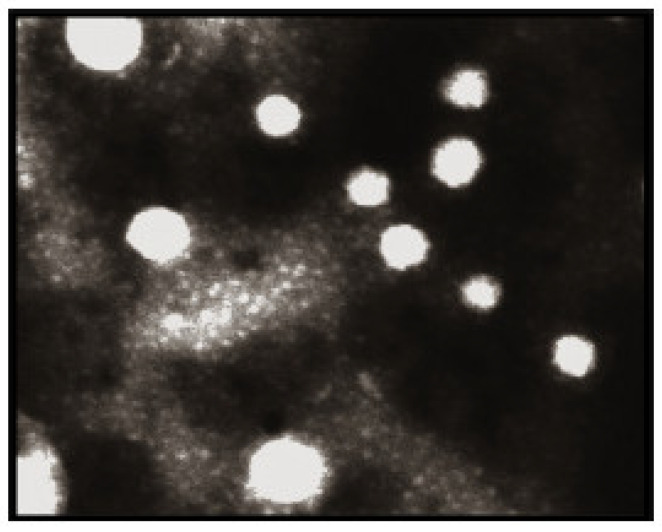
Quercetin-loaded nanoemulsion globules’ morphology was observed using TEM at a voltage of 60–80 kV (magnification 10,000, scale 200 nm) [43]. “Reproduced with permission from reference”.

**Figure 7 molecules-28-02540-f007:**
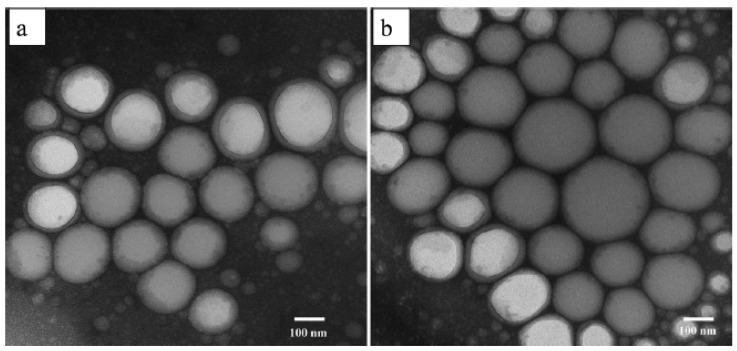
TEM images of the best nanoemulsions without quercetin (**a**); TEM images of the best Q-SNEDDS nanoemulsions at 72 h after Q-SNEDDS dilution (**b**) [44]. “Reproduced with permission from reference”.

**Figure 8 molecules-28-02540-f008:**
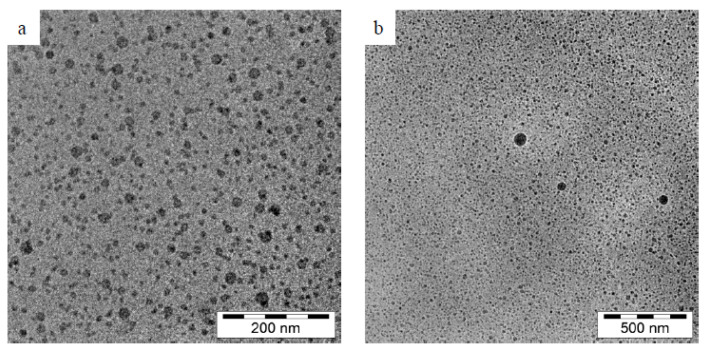
Bioactive cationic nanoemulsion is depicted in two cryo-TEM images: (**a**) at 200 nm and (**b**) at 500 nm [45]. “Reproduced with permission from reference”.

**Figure 9 molecules-28-02540-f009:**
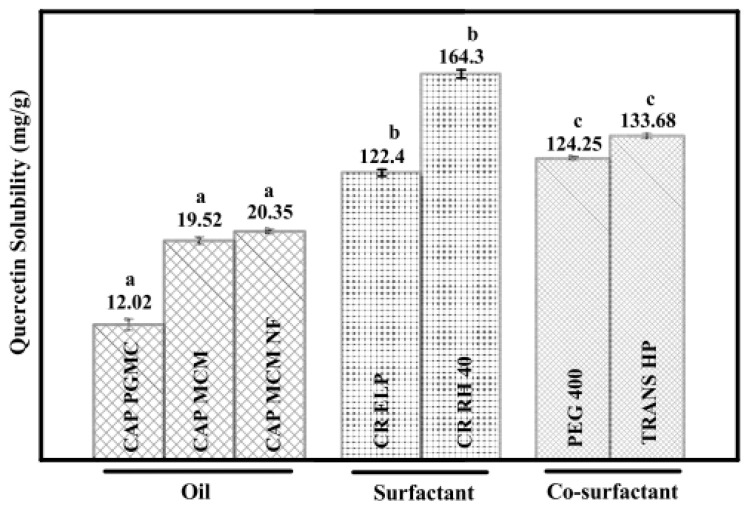
Solubility of quercetin in oils, surfactants, and cosurfactants. Different lower-case letters between the groups (oil, surfactant, and co-surfactant) are significantly different from each other (*p* < 0.05, *n* = 3, error bars show standard deviation) [46].

**Figure 10 molecules-28-02540-f010:**
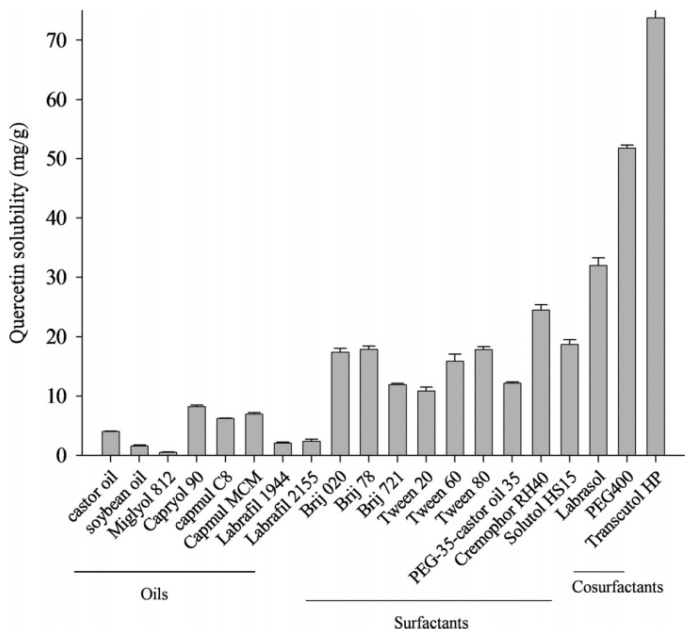
Quercetin solubility in a variety of oils, surfactants, and cosurfactants [44].

**Figure 11 molecules-28-02540-f011:**
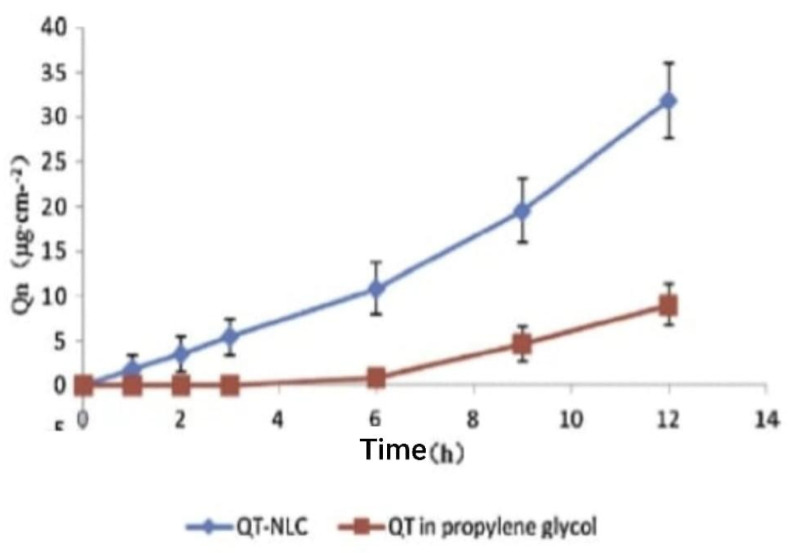
Profiles of QT permeation through excised rat skins from QT propylene glycol solution and QT-NLCs [50].

**Table 1 molecules-28-02540-t001:** Relative Diffusion Coefficient Values of Water in a Nanoemulsion Loaded with Quercetin at 10% Emulsifier Content and Calculated from NMR Results at 25 °C [2].

Oil Concn (%)	D_W_/D^W^_0_	Oil Concn (%)	D_W_/D^W^_0_
5	0.752	30	0.569
10	0.657	40	0.558
20	0.512		

**Table 2 molecules-28-02540-t002:** FTIR frequency ranges and functional groups present in QUE.

Presence	cm^−1^	References
C=O stretching	1660	[46,47,48,49]
–OH stretching	3397	[46,47,48,49]
–CH stretching	2717	[46,47,48,49]
-C-O-C stretching	1111	[46,47,48,49]
= CH bending	830	[46,47,48,49]

**Table 3 molecules-28-02540-t003:** Drug content, recovery, and effectiveness of entrapment [52].

nEs	Content of +Curcumin (mg/mL)	Recovery of Curcumin (%)	Content of Quercetin (mg/mL)	Recovery of Quercetin (%)	Effective of Entrapment (%)
CQ_NE-	0.61 ± 0.01	81.33 ± 1.30	0.72 ± 0.01	96.00 ± 1.30	>99
CQ_NE+	0.62 ± 0.02	82.66 ± 2.60	0.71 ± 0.03	94.66 ± 4.00	>99
CQ_NEgel	0.61 ± 0.01	81.33 ± 1.30	0.72 ± 0.01	96.00 ± 1.30	>99

## Data Availability

Not applicable.

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
