# Peer review of "Quantification of the Quercetin Nanoemulsion Technique Using Various Parameters"

_molecules, 2023, doi:10.3390/molecules28062540_

Round 1
Reviewer 1 Report
To the Authors of the manuscript ‘Exploring the trending Research in Quercetin’.
I have thoroughly reviewed the subject matter of the manuscript and I am satisfied. The work summarizes an interesting and important topic (this summary should be emphasized in the Abstract! in stead of studies..), sufficient to meet the interest for broad readership. However, I regret to say that there are some minor examples of negligence. I pointed out my observations below.
- I suggest the reorganization of the text. Please reorder the Introduction due to some repetitions... Additionally, in the line 38 you mentioned poor solubility of quercetin and it average daily intake in the line 32... I believe that some comment about quercetin in the forms of an aglycone as well as glycosides is needed
- In my opinion, section 2 should be devoted to the explanation what micro/nanoemulsions are (you use these names interchangeably which is confusing to the reader. Please give a brief explanation) -> then effect of external parameters on quercetin nanoemulsion can be presented. I also suggest to consider the addition of the section about nanoparticles.
- The section 3 may be about the techniques/methods used in the characterisation of quercetin nanoemulsion. In this section, I suggest a brief introduction to each technique (the current summary), and then citing examples from the literature.
- In my opinion the section Results and Discussion is misleading and should be combined with the Conclusion section
- Editorial corrections are required in the whole text (italic script, e.g. in vitro; UV-Vis or UV-VIS; please correct "pH").
- The list of authors should be corrected
- Please check the use of abbreviations, enter an abbreviation the first time you use it and then continue to use it (you also need to update the list of abbreviations). Please check the whole manuscript.
- The titles of Figure 5 and 8 (Fig. 8d?) needs to be improved.
- The quality of Figures 9 and 11 needs to be improved.
- I recommend to consult the manuscript with a natvie speaker. Please see the lines 160-162, 170, 211-212, 245, 266, 290, 318-319.
- In the line 157: "Standard solutions (2.5 g/mL)..." - is the unit correct? See also the line 188 and 298.
- In the line 161: please explain "QU". It is QUE, isn't it?
- The paragraphs 279-289 and 302-305 should be in the Introduction.
Author Response
Response to Reviewer 1 Comments
I suggest the reorganization of the text. Please reorder the Introduction due to some repetitions... Additionally, in the line 38 you mentioned poor solubility of quercetin and it average daily intake in the line 32... I believe that some comment about quercetin in the forms of an aglycone as well as glycosides is needed
Response: Thanks, we have removed the repeated text and put our comments about quercetin in the forms of an aglycone as well as glycosides.
In my opinion, section 2 should be devoted to the explanation what micro/nanoemulsions are (you use these names interchangeably which is confusing to the reader. Please give a brief explanation) -> then effect of external parameters on quercetin nanoemulsion can be presented. I also suggest to consider the addition of the section about nanoparticles.
Response: Thanks for suggestion. We gave some explanation of micro/nanoemulsions to clarify the difference between these two. After that we gave general introduction of nanoparticles on line number 56-77 of page 2. Following your suggestion, we placed effect of external parameters on quercetin nanoemulsions accordingly.
The section 3 may be about the techniques/methods used in the characterisation of quercetin nanoemulsion. In this section, I suggest a brief introduction to each technique (the current summary), and then citing examples from the literature.
Response: We briefly described each technique with methodology on line number 372 -378 (pg no. 10-11), line number 296-305 (pg no. 8), line number 324-330 (pg no. 8-9), line number 441-447 (pg no. 13), line number 381-385 (pg no. 11).
In my opinion the section Results and Discussion is misleading and should be combined with the Conclusion section
Response: Corrections have been done in the conclusion section.
Editorial corrections are required in the whole text (italic script, e.g. in vitro; UV-Vis or UV-VIS; please correct "pH").
Response: We removed all these errors.
The list of authors should be corrected
Response: Corrections have been done in the revised manuscript.
Please check the use of abbreviations, enter an abbreviation the first time you use it and then continue to use it (you also need to update the list of abbreviations). Please check the whole manuscript.
Response: Corrections have been done in the revised manuscript on line number 515-525 of pg 15.
The titles of Figure 5 and 8 (Fig. 8d?) needs to be improved.
Response: We have improved Figures 5 and 8 (Fig. 8d) which presented in Figure 9 & 1.
The quality of Figures 9 and 11 needs to be improved in Figure 2 & 4
Response: We have improved Figures 9 and 11 which presented in Figure 2 & 4.
I recommend to consult the manuscript with a natvie speaker. Please see the lines 160-162, 170, 211-212, 245, 266, 290, 318-319.
Response: We have consulted the native speaker and corrected the revised manuscript.
In the line 157: "Standard solutions (2.5 g/mL)..." - is the unit correct? See also the line 188 and 298.
Response: g/mL is the correct unit mentioned in line number 444 (pg no. 13).
In the line 161: please explain "QU". It is QUE, isn't it?
Response: We have changed QU into QUE.
The paragraphs 279-289 and 302-305 should be in the Introduction.
Response: We have adjusted in the introduction part.

Reviewer 2 Report
This paper is considered to be a result that can be applied to cosmetics, food, and pharmaceutical fields in the future as a nanoemulsion technology. However, the composition of Manuscript is not smooth. We suggest that the method part of this research paper be written transparently, and the results are also hoped to be supplemented.
1) Consider a title change; The technique of quercetin nanoemulsion technique by various parameters
2) The author should consider rewriting by dividing the methods and results from the 2. Characterization of Quercetin Nanoemulsion section.
Author Response
The review depicts the important role of the quercetin nanoemulsion technique, used to enhance the solubility of target materials both in vivo and in vitro as well as to decrease the risk of degradation and metabolism of drugs. The surface morphology of quercetin nanoemulsion has been studied using cryo-TEM images. The effect of different parameters like pH, salts, and solvent concentration on quercetin nanoemulsion has been studied. UV-vis spectroscopy and HPLC were used to determine various properties of quercetin nanoemulsions, like their solubility, stability, and encapsulating efficiency.
However, in my point of view, the information in the review can be extended with other trends used in modern chemistry, concerning characterization of quercetin nanoemulsion, for example, and I have reviewed articles about NMR (nuclear magnetic resonance) or FITR.
Response: Thank you for indicating the gap. We have addressed it at the subtopic above and explained quercetin nanoemulsions behavior in light of FTIR and NMR which has explained on line number 372 -378 (pg no. 10-11), line number 296-305(pg no. 8).
Consider a title change; The technique of quercetin nanoemulsion technique by various parameters
Response: We have changed the title.
The author should consider rewriting by dividing the methods and results from the 2. Characterization of Quercetin Nanoemulsion section.
Response: Corrections have been done in the revised manuscript which has explained on line number 372 -378 (pg no. 10-11), line number 296-305 (pg no. 8), line number 324-330 (pg no. 8-9), line number 441-447(pg no. 13), line number 381-385 (pg no. 11).

Reviewer 3 Report
The review depicts the important role of the quercetin nanoemulsion technique, used to enhance the solubility of target materials both in vivo and in vitro as well as to decrease the risk of degradation and metabolism of drugs. The surface morphology of quercetin nanoemulsion has been studied using cryo-TEM images. The effect of different parameters like pH, salts, and solvent concentration on quercetin nanoemulsion has been studied. UV-vis spectroscopy and HPLC were used to determine various properties of quercetin nanoemulsions, like their solubility, stability, and encapsulating efficienc.
However, in my point of view, the information in the review can be extended with other trends used in modern chemistry, concerning characterization of quercetin nanoemulsion, for example, and I have reviewed articles about NMR (nuclear magnetic resonance) or FITR.
Improve the quality of fig 5,6,7 and 9.
Author Response
The review depicts the important role of the quercetin nanoemulsion technique, used to enhance the solubility of target materials both in vivo and in vitro as well as to decrease the risk of degradation and metabolism of drugs. The surface morphology of quercetin nanoemulsion has been studied using cryo-TEM images. The effect of different parameters like pH, salts, and solvent concentration on quercetin nanoemulsion has been studied. UV-vis spectroscopy and HPLC were used to determine various properties of quercetin nanoemulsions, like their solubility, stability, and encapsulating efficiency.
However, in my point of view, the information in the review can be extended with other trends used in modern chemistry, concerning characterization of quercetin nanoemulsion, for example, and I have reviewed articles about NMR (nuclear magnetic resonance) or FITR.
Response: Thank you for indicating the gap. We have addressed it at the subtopic above and explained quercetin nanoemulsions behavior in light of FTIR and NMR which has explained on line number 372 -378 (pg no. 10-11), line number 296-305 (pg no. 8).
Improve the quality of fig 5,6,7 and 9.
Response: We have improved the quality of fig 5,6,7 and 9 which has presented as
Figure 9, 10, 11, 2 respectively.

Round 2
Reviewer 2 Report
The author appreciates the correction of the title and sentence content. However, if the methods and results are divided, it will be a good research paper.